# An mHealth tool for community health workers to improve caregiver knowledge of child health in the Amazon: An effectiveness-implementation hybrid evaluation

**Christopher M. Westgard** [1,2]*, **Luis A. Orrego-Ferreyros** [1,3]

**1** Department of Research and Innovation, Elementos, Lima, Peru, **2** Department of Maternal and Child Health, Gillings School of Global Public Health, University of North Carolina at Chapel Hill, Chapel Hill, North Carolina, United States of America, **3** Facultad de Medicina Alberto Hurtado, Universidad Peruana Cayetano Heredia, Lima, Peru

* cmwestgard@gmail.com

**Data Availability Statement:** The dataset supporting the conclusions of this article is

## Abstract

When community health workers (CHWs) are effective, they can teach healthy child rearing practices in their communities and improve child health and development outcomes. An effective mHealth tool can improve the capacity of CHWs to transmit knowledge to caregivers. This article evaluates the implementation of an mHealth tool in a CHW program in the Amazon of Peru. The intervention was designed, implemented, and evaluated with the guidance of multiple implementation science tools. A Hybrid Type 3 evaluation design was used to test the effectiveness of the implementation strategies and appropriateness of the intervention. The implementation outcomes: acceptability, adoption, dosage, and fidelity were analyzed with mixed methods approach to determine if the intervention was successfully installed in the CHW program. The service outcome, knowledge scores, was analyzed with an independent samples t-test and one way ANOVA to determine the effect of the program. The implementation strategies resulted in high degrees of acceptability, adoption, and fidelity of the mHealth tool. The surveillance component of the mHealth tools was not adequately adopted. The group of caregivers that received home visits with the mHealth tool (N = 48) had significantly higher knowledge scores (+1.26 standard deviations) than those in the control group (N = 138) ($t$(184) = -4.39, $p$<0.001). The COVID-19 pandemic significantly decreased the dosage of the intervention received by the participants. The CHEST App intervention is a promising tool to improve the capacity of CHWs during their home visits. Trial registered with ISRCTN on 11/29/2018 at https://doi.org/10.1186/ISRCTN43591826.

## Background

Children around the world continue to suffer from ailments such as malnutrition and developmental delay due to unhealthy practices in the household [1–4]. In the Amazon region of Peru, in the department of Loreto (2020), 69% of children under 3 have anemia, 31% under 5 have

available in the figshare repository, https://figshare.com/articles/dataset/Implementation_Research_for_the_Evaluation_of_the_Child_Health_Education_and_Surveillance_Tool_Application/13681423.

**Funding:** Project undertaken with the financial support of a grant from Saving Brains (#1809-18532 to CMW), Grand Challenges Canada and the Government of Canada, through Global Affairs Canada (GAC). All authors were supported from the Saving Brains grant. The funders had no role in study design, data collection and analysis, decision to publish, or preparation of the manuscript.

**Competing interests:** The authors have declared that no competing interests exist.

**Abbreviations:** C.A.R.E Guidelines, Consolidated Advice on Reporting ECD Implementation Research Guidelines; CHEST App, Child Health Education and Surveillance Tool Application; ECD, Early Childhood Development; ICT, Information and Communication Technology; IR Logic Model, Implementation Research Logic Model.

chronic malnutrition, and 26.7% of children are developmentally delayed [5,6]. Although poverty constrains caregivers' ability to adopt certain healthy practices in the household, a large portion of ailments can be alleviated by the adoption of better sanitation practices, diet, and disease prevention [7–9]. Unfortunately, caregivers in impoverished settings often have incomplete knowledge of healthy practices due to poor education and health promotion efforts [7,10–14]. For example, one study in the amazon found that 30% of people interviewed did not understand their own chronic illness diagnosis [10]. Another study in the same region found that community health workers (CHWs) had limited knowledge about how to prevent and treat diarrhea [13].

Health education and promotion has been difficult to achieve at scale in rural communities, but when accomplished it can substantially improve child health and development [15,16]. The utilization of CHWs to conduct health promotion and education for child health and development has been shown to be effective, though outcomes vary greatly [16–19]. Effective, consistent, and scalable implementation of CHWs programs have been elusive [13,16]. The impact of the CHW programs is hindered by poor performance of CHWs, a lack of effective educational material, and poor structure to guide home visits [13,20]. The CHWs are often unequipped and under-trained to successfully transmit the information to the caregivers and convince them the importance of the behavior change [13,20,21].

The utilization of information and communication technology (ICT) and mHealth technology have been shown to be effective at supporting CHW programs and improving child growth and development [22–30]. Frontline workers have used mHealth tools for a variety of purposes, including patient management, work planning and scheduling, education and awareness, clinical support, performance management, and information systems support [31]. Although the tools have shown promise, effective implementation of mHealth tools in rural settings has proven to be difficult to achieve and sustain [30]. Most populations in impoverished settings have yet to receive the benefits of modern ICT to support CHWs [32].

Ineffective implementation of evidence-based interventions, such as CHW programs and mHealth tools for behavior change, is partly due to an imprecise understanding of what has been done, what has worked, and what has failed [33]. The implementation of innovations needs to be mapped, evaluated, and reported with sufficient detail to support continuous learning and improvement [34–36]. To help fill the knowledge gap, the current study reports on the implementation and intermediary outcomes of an mHealth innovation to support CHWs in the Amazon of Peru. The study reports on the effectiveness of implementation by evaluating the implementation outcomes, and the effectiveness of the intervention at improving CHW performance by evaluating changes in knowledge scores by caregivers. The implementation research will help generate data on what works and what does not when using mHealth tools to support CHW programs in rural settings.

## Methods

### Research ethics approval and consent to participate

The study protocol was approved by the national ethics committee in Peru at the "Hospital Nacional Docente Madre Niño "San Bartolome,"" on November 8, 2018 (Exp. Number 15 463–18, Oficio N. 0744–2018- OADI- HONADOMANI- SB) and the Institutional Review Board, Office of Human Research Ethics at the University of North Carolina—Chapel Hill (IRB Number: 19–3097). The participants provided their written informed consent to participate in this study.

## Participants and setting

The study took place in Peru, department of Loreto, districts of Indiana and Fernando Lores. The intervention community (Indiana) and the control community (Tamshiyacu) are similar in size (8–10,000), access to health facility (Level I-2 in community), distance to province capital (1.5 hours in fast boat), rates of anemia (25–32%), and structure of CHW program. Both communities had an established CHW program in their community before the intervention began, with nearly identical program structure. To determine if the intervention and control groups were similar at baseline, an independent t-test analysis was conducted with their knowledge scores.

All participants provided written consent to participate in the study before receiving any survey or intervention activities. All participants were over the age of 18. Caregivers provided all information related to the children in the study. A detailed description of the selection process and study population is included in the published study protocol [37]. The trial was registered with ISRCTN on 11/29/2018 at https://doi.org/10.1186/ISRCTN43591826.

The pilot study was interrupted by the COVID-19 pandemic, causing CHW program shutdowns and thus a significant loss of program participants and decrease in home visits conducted. After 6 months of shutdowns, both communities re-activated their CHW program with new precautions, such as face shields and masks, and meeting outside of the house.

## The intervention

In the communities, CHWs conduct home visits with new mothers to teach them health topics such as sanitation, diet, disease prevention, and early childhood development. To address underperformance of the CHW program, the research team at the Peruvian research organization, Elementos, developed a tablet-based application, home visit curriculum, and animated videos to support the CHWs. The application was built to help teach caregivers key health messages and collect child health indicators during home visits. The mHealth tool is titled, The Child Health Education and Surveillance Tool Application (The CHEST App). The CHEST App selects the health messages and animated videos to present during the home visit based on the age of the child, it calculates anthropometric outcomes based on heigh, weight, and age of the child, it provides a caseload screen to display the health status of the child and next scheduled home visit, and it uploads the data collected during the home visit to a server. A video of the CHEST App and the animated videos that accompany the App can be viewed online [38]. The supervisors of the CHWs use the App to monitor the frequency of home visits and the health status of the children. The primary objectives of the CHEST App intervention are to improve early childhood development (ECD) scores, reduce anemia rates, and reduce chronic malnutrition rates. Further details about the CHW program and intervention, as well as the activities conducted to plan and install the program during each stage of implementation (based on the Active Implementation Frameworks [39,40]) are described in the article, Westgard, et. al., 2020 [41]. The theory of change of the intervention is displayed in S1 Fig and the Conceptual Model of the intervention is displayed in S2 Fig. The implementation strategies to install the intervention included training and coaching, identifying and preparing champions, assessing readiness and identify barriers, continuous improvement cycles for adaptation, capture and share local knowledge, getting by-in from local opinion leaders, and others. The full list of implementation strategies is described in detail in the article Westgard, et. al., 2020 [41]. The current study aims to evaluate the implementation of the CHEST App intervention and assess the impact of the intervention on the intermediary outcome, knowledge of healthy practices by caregivers. The evaluation will help determine if the innovation should be scaled or replicated, and to identify opportunities to improve implementation strategies during scale-up

and replication in other contexts. An impact evaluation to assess changes in child health and developed will be conducted in the future.

## Study design

The implementation process and CHEST App intervention were evaluated using a Hybrid Type 3 evaluation study design [42]. In a Hybrid Type 3 study, the implementation strategies are evaluated to determine if the intervention was successfully installed in the local context and created patient-level changes [42]. The primary outcome of the study knowledge of healthy childrearing practices by caregivers. The secondary outcomes are the implementation outcomes, including, acceptability, adoption, fidelity, and dosage.

The two communities involved were randomly assigned to the intervention group or control group. The intervention group's CHWs received the CHEST App and CHEST App training, while the control group's CHWs did not. For the analysis, the intervention group included the caregivers that received a home visit with the CHEST App. The control group included all children in both communities that did not receive the CHEST App. A baseline survey was conducted prior to implementation and an endline survey was conducted 16 months following implementation, in the intervention and control communities.

## Service outcome

The knowledge score represents the service outcome of the study, as shown in IR Logic Model (S3 Fig). The service outcome of the study reflects the effectiveness of the intervention to improve CHW performance [43]. If CHWs are effective at delivering their intended service, the caregivers will have greater knowledge of healthy childrearing practices. Improved knowledge is expected to lead to improvements in clinical outcomes (anemia and ECD scores) following higher levels of dosage [44]. Knowledge scores were measured by an opened-end questionnaire with caregivers, at baseline and endline. The questionnaire was designed to give the participant ample opportunity to describe what they know about each question. The surveyors were trained to ask the participant the survey question, then probe the participant to provide further information. For example, the participant was asked, what are the benefits of breast milk for a baby, then follow-up with statements such as, "what other benefits" and "have you heard of any other benefits". Probes continued until the participate indicated that they do not know any further information. For each correct answer, the participant received a point. The points were totaled to provide the knowledge score for the participant. The questionnaire included 15 questions with a total possible score of 91. The topics included nutrition, sanitation and hygiene, disease prevention, and early childhood development.

## Implementation outcomes

Evaluation of the implementation outcomes was conducted with a mixed methods approach to identify the extent to which the intervention was successfully incorporated into the local CHW program. The implementation research reported here follows the C.A.R.E. guidelines (Consolidated Advice on Reporting ECD Implementation Research) to ensure the necessary information is included in the evaluation [33]. An Implementation Research Logic Model (IR Logic Model) was used to design the evaluation process and specify the relationship between the determinants of implementation, implementation strategies, and the implementation and service outcomes (S3 Fig) [34]. The implementation outcomes were selected based on the conceptual framework presented by Proctor, et. al., 2011 [43]. The implementation outcomes serve as a precondition for attaining the desired changes in service and clinical outcomes. The implementation outcomes relevant to this study include, acceptability, adoption, and fidelity.

Other implementation outcomes were not measured because the intervention had not yet reached the point in its maturation to measure them with confidence, including, feasibility, penetration, and sustainability. An additional implementation outcome, dosage, was also measured.

Acceptability was measured to identify the extent to which the implementation stakeholders (caregivers and providers) perceive the intervention to be satisfactory [43]. Both the CHWs and caregivers in the intervention group received a semi-structured interview to determine their level of acceptance of the CHEST App. The quantitative portion of the survey was analyzed and reported using descriptive statistics. The qualitative responses were analyzed by identifying a set of sub-themes from the responses then assessed for similarities and differences in perspectives of the participants. Key quotes that best reflect the position of each actor group are reported.

Adoption was measured to determine the uptake of the intervention and continued use by the providers throughout the study period. Adoption was used to indicate if the CHW used the CHEST App consistently during their home visits and if the program coordinators used the CHEST App for surveillance purposes.

Adoption by CHWs was measured by analyzing data that was collected with the CHEST App and then uploaded to the server. The number of home visits conducted with the CHEST App was compared to the number of home visits the CHWs were expected to complete each month, to determine a percentage of adoption. Also, the CHWs received a semi-structured interview to describe the extent to which they use the CHEST App during their home visits, and the components of the CHEST App they use with consistency. The program coordinator of the CHW program received a semi-structured survey pertaining to the adoption of the CHEST App and the use of each of its components. The responses to the survey are summarized.

Dosage is a measure of the total number of home visits realized with the CHEST App during the study period. Dosage is measured to determine to what extent the intervention reached the recipients.

Fidelity was measured to determine the degree to which the intervention was delivered as prescribed. The research team conducted observations of home visits with the CHEST App to determine if each component of the CHEST App was being delivered with quality and as described in the intervention protocol. The observers had an observation checklist to mark the completion, or incompletion, of each component. The results of the checklist are reported as the measure of fidelity.

A SPIRIT Checklist was completed to ensure the manuscript includes all research reporting components. The SPIRIT Checklist is included in S4 Fig. A CONSORT Flow Diagram to describe recruitment is included in S5 Fig.

## Results

The results of the implementation research can be seen in the IR Logic Model in S3 Fig. The evaluation included 48 caregivers and 6 CHWs that received the intervention and 138 caregivers in the control group at endline.

### Knowledge scores by caregivers

The average knowledge scores by the caregivers, at baseline and endline, are shown in Table 1. The scores were normally distributed within each group. The intervention and control group showed no significant difference in knowledge scores at baseline (p = 0.9216). The independent samples t-test (intention to treat) found a significant increase in knowledge scores

**Table 1. Mean knowledge scores.**

| Knowledge Scores | Baseline | Endline |
|---|---|---|
| Intervention Group | 23.25 (N = 47) | 33.19 (N = 48) |
| Control Group | 23.4 (N = 72) | 27.66 (N = 138) |

(M = 5.53, SD = 1.26) by those in the intervention group (N = 48) compared to those in the control group (N = 138) ($t(184)$ = -4.39, $p<0.001$). The results of the independent t-test analysis can be seen in Table 2. A one way ANOVA showed that the effect of the CHEST App on knowledge scores yielded significant variation among groups, $F(1,185)$ = 12.9, $p<0.000$. The results of the one-way ANOVA analysis can be seen in Table 3. A post hoc Tukey test indicated that the average knowledge score for those that received 1–2 home visits (M = 36.05, SD = 6.11) was significantly higher than those that received no home visits (M = 27.66, SD = 7.38, $p<0.000$). The average knowledge score of those that received 3+ home visits was significantly different than those that received 1–2 home visits, (M = 30.77, SD = 8.57, $p$ = 0.040). The comparison between those that received no home visits and those that received 3+ home visits was not significant, $p$ = 0.125. The results of the post hoc Tukey test can be seen in Table 4.

## Acceptability by caregivers

Most caregivers (92%, n = 45) that received home visits with the CHEST App expressed that they prefer the App than use of traditional methods (pen, paper, and flipcharts). 5% (n = 2 expressed that they had no preference and 3% (n = 1) expressed that they prefer traditional methods of home visits.

When caregivers were asked if they learned the health messages delivered during the home visits better with the CHEST App or without the CHEST App, 84% (n = 41) expressed that they learned better with the CHEST App, 13% (n = 6) expressed that they learned the same with or without it, and 3% (n = 1) expressed that they learned better with traditional methods. When asked what they liked most about receiving home visits with the CHEST App, all caregivers answered that they most enjoyed the educational component of the App. Many (43%, n = 21) specifically mentioned the animated videos as a reason why they prefer the CHEST App to traditional methods. For example, a mother in the community of Indiana said,

*"I like that they show use the videos and how to feed the children. You learn better because you can see how to do it through the animations"*.

**Table 2. Independent t-test analysis results to compare group means of knowledge scores.**

| Outcome | Intervention Group | | Control Group | | P value |
|---|---|---|---|---|---|
| | Sample size | Mean | Sample size | Mean | |
| Knowledge Scores | 48 | 33.19 | 138 | 27.66 | <0.000 |

**Table 3. One-way ANOVA results to compare group means of knowledge scores.**

| Outcome | No home visits | 1–2 Home Visits | 3+ Home Visits | Sum of Squares | Df | F | P value |
|---|---|---|---|---|---|---|---|
| | | Mean (N) | | | | | |
| Knowledge Scores | 27.66 (138) | 36.05 (22) | 30.77 (26) | 1420.06 | 2,183 | 12.9 | <0.000 |

**Table 4. Tukey Post hoc comparison of groups on knowledge scores.**

| Groups | Mean Difference | P value |
|---|---|---|
| No home visits vs 1–2 home visits | 8.39 | <0.000 |
| No home visits vs 3+ home visits | 3.11 | 0.125 |
| 1–2 home visits vs 3+ home visits | -5.28 | 0.040 |

*(Me gusta que nos hacen mirar los videos y la alimentación de los niños. Se aprende mejor porque se ven como se hace para aprender mediante dibujos)*

## Acceptability by CHWs

All CHWs expressed that they prefer to conduct the home visits with the CHEST App than their traditional methods (pen, paper, and flipcharts). When asked what they liked about using the App, the CHWs reported that they most liked learning from the educational material, the animated videos, and the child health indicators displayed in red or blue. The animated videos were the most cited reason that they liked the CHEST App and how it helped them better conduct their home visits. A CHW in the community of Indiana said,

> "*Yes, it (the App) has everything summarized and is faster. They like the videos. They laugh and understand more quickly. The child points. They like it a lot.*"

> *(Si. Tiene todo resumido y mas rápido. Les gustan los videos. Se ríen. Aprenden más rápido y el niño apunta. Les gusta mucho)*

The CHWs expressed that the most difficult part about using the CHEST App was sending the data and registering the information in the tablet.

## Adoption by CHWs

The CHWs were registering health indicators from 85 home visits per month with the CHEST App, the same number of visits they were assigned. Thus, adoption was confirmed to be 100%, meaning the App was used by the CHWs during every scheduled home visit. After 18 months of use, the CHWs expressed that they continue to use the CHEST App during all their home visits. The caregivers verified adoption of the CHEST App by CHWs by reporting the number of home visits they have received, and number of home visits received with an electronic tablet. The results were similar (8.6 visits vs. 7.8 visits with tablet), indicating that the CHWs use the CHEST App during their home visits. All the CHWs reported that they use all the App's functions when conducting a home visit (share health indicators with caregiver, register health indicators, scheduling, educational images, and animated videos).

Adoption of the CHEST App was hindered due to the cancelation of the CHW programs in several communities, both before COVID and during COVID. There were several communities that indicated that they had a CHW program and wanted to receive the CHEST App intervention, however, by the time the App was ready for implementation the CHW program had discontinued. The instability of the local CHW programs makes adoption and sustainability of the CHEST App difficult to measure because without the infrastructure of the CHW program the CHEST App cannot be utilized.

## Adoption by CHW program coordinators

CHW program coordinators were tasked to upload and utilize the data collected with the CHEST App. Adoption of this practice was not achieved. The program coordinators were not

interested in uploading the data to the server. The task was completed during the pilot study, but only to satisfy the external research team at Elementos. Therefore, the research team is not able to monitor the data from the tablet unless they visit the community and upload the data from the tablets to the server. The program coordinator and local municipality continue to desire a paper-based list of results of the surveillance data. The program coordinator wrote the data displayed in the tablet on paper and submitted the paper report to the municipality. Thus, reflecting low adoption and poor fidelity of the surveillance function of the CHEST App.

## Dosage

A total of 140 children received home visits with the CHEST App. The CHWs conducted a total of 686 home visits while the program was operating. The program fell significantly short of reaching its dosage objective of 1200 home visits, at 57% of the objective. It was estimated that 1200 home visits with the CHEST App were needed to create significant change in child health outcomes. The dosage drop was due to cancellations and layoffs due to the COVID-19 pandemic.

The intervention group community continues to use the CHEST App in their program at the time of writing this manuscript, 26 months post implementation.

## Fidelity

Fidelity of the CHEST App by the CHWs, as observed during home visits, remained high throughout the pilot. At 18 months, each intended activity associated the CHEST APP showed fidelity scores between 80–100%, as shown in Table 5. The score indicates a high degree of fidelity and high quality of home visits with the CHEST App.

## Discussion

The results suggest that the CHEST App was successfully implemented and improved the performance of CHWs to teach caregivers important knowledge pertaining to healthy child rearing. The caregivers that received visits from CHWs that utilized the CHEST App displayed greater knowledge (1.26 standard deviation increase) than caregivers that received visits from CHWs with traditional methods (no material or flip charts and brochures). Improvement in knowledge scores by the caregivers reflects the immediate impact of the CHEST App. Similar to previous studies, the mHealth tool improved the ability of the CHWs to conduct health promotion and education in the household [27]. Improved knowledge by the caregivers is expected to contribute to improve child health and development of their children [7–9,23,45].

Programs that show high levels of acceptance, adoption, and fidelity of the intervention, as well as positive improvements in intermediary/patient-level outcomes, are likely to be effective at changing clinical level outcomes, such as early childhood development of the program recipients [31,43,46,47]. Although the implementation outcomes and service outcomes of the

**Table 5. Fidelity scores for the CHEST App.**

| Intended Activities with CHEST App | Fidelity Score (N = 6) |
|---|---|
| Registered Health Indicators | 80% |
| Used App to discuss child health indicators with caregiver | 80% |
| Used content in App to explain health messages | 100% |
| Explained health messages with sufficient information | 100% |
| Asked caregiver what they understood from the health messages | 100% |
| Showed animated video to caregiver | 80% |

current study showed effectiveness, the final impact of the CHEST App intervention on the clinical outcomes (such as anemia, chronic malnutrition, and early childhood development) will need to be assessed after the children have received a greater number of home visits with the CHEST App.

Prior to the implementation of the intervention, applied implementation research was conducted to identify effective strategies to install the intervention in the local context. The research team first assessed the key implementation determinants in the local context to identify potential barriers and facilitators to success. The determinants were based on the categorization from the Consolidated Framework for Implementation Research (CFIR), Damschroder et al., 2009 [48]. To help determine which implementation strategies should be included, the research team used the CFIR ERIC Matching Tool [49]. The strategies that were used during the implementation process are listed in the IR Logic Model. Not all strategies suggested by the Matching Tool were utilized. For instance, funding strategies were not used due to a lack of receptibility by the local government to discuss changes to funding structures. For each implementation strategy, the research team implemented a set of discrete activities to operationalize the strategies [50,51]. Methods from Intervention Mapping (and Implementation Mapping) were used to develop the implementation strategies and select indicators to evaluate to measure the implementation outcomes [52]. The specific implementation activities are described in detail in Westgard, et al., 2020 [41].

The implementation outcomes that were evaluated suggest that the implementation strategies were mostly effective at installing the intervention in the established CHW program, with high degrees of acceptability, adoption, and fidelity. Acceptability and adoption by the CHWs were greatly influenced by the novelty of the CHEST App technology, as similar expressed in previous research [22,28,53]. The CHWs reported feeling empowered from using the modern technology and more prepared to teach the caregivers with the extensive educational material they had at hand. Fidelity rates of the CHEST App by the CHWs were consistently high, which is especially promising given the dynamic nature of home visits. Registering the child health indicators with the CHEST App was the most likely to be omitted during the home visit. For instance, if a home visit occurred and the caregiver did not have new information written on their child's growth monitoring card, the CHWs would not register child health indicators during that visit. This led to missed opportunity to record cases of infections that may have occurred since the last home visit. All other components of the CHEST App were used with consistency and according to protocol. The high level of adoption and fidelity suggests that the caregivers received the intervention as intended, which has been shown in previous studies to be an essential element for improved quality of services by CHWs [23,45].

Full adoption and fidelity by CHW program coordinators were not achieved because there was little desire to digitize the surveillance data. The intervention, for the program coordinators, included uploading data from the tablets to a server and using the data to track the health of the child in their communities and make data-informed decisions. The coordinators and local municipality members did not have the desire to rely on the technology to track the health of children in their communities. Because the programs are not integrated into a larger, regional program, the number of children managed by the program coordinators is relatively small. The program coordinators expressed that they can more easily manage the information of the children with their previous methods (pen and paper) than work with the server and navigate the challenges of having internet connection. The CHW program coordinators value the educational component of the CHEST App but do not value the digital surveillance component. The digital surveillance component will be more important for CHW programs that are integrated into a regional or national level program, which demands greater management of data.

Implementation outcomes related to sustainability and penetration need to be assessed after the communities have had more time with the CHEST App, without the support from the external research team from Elementos. Sustainability of the CHEST App intervention is greatly influenced by the sustainability of the CHW programs. The local municipalities have been inconsistent in their support of the CHW programs. Multiple municipalities cancelled or significantly changed their CHW program during the program planning phase of the CHEST App intervention. The two municipalities in the study temporarily canceled their CHW program due to COVID and a lack of available funds. The challenge of sustaining CHW programs is well documented and remains to be a significant influence in their under-performance. CHW programs suffer from lack of resources and short-term funding, program disruptions due to political changes, high levels of attrition, and wavering supervision/accountability [18,54–56]. The current study similarly suffered from program disruptions and decrease in number of home visits. The political decision to include the CHEST App in a CHW program should include consideration regarding the stability and sustainability of the program. The tablet represents a significant extra cost and would only create enough value for the recipients to outweigh the cost if the program is sustained for long-term. To determine the long-term impact of the CHEST App on the health and development of the children a further effectiveness trial is needed, without stoppage or decrease in doses due to external factors. The authors recommend that the CHEST App be integrated and piloted as part of the National CHW program, Cuna Mas [57], that conducts home visits throughout the poorest regions of Peru in a more stable, sustainable way.

## Conclusions

The effectiveness-implementation hybrid type 3 study provided the following results:

1. The CHEST App intervention can be effectively installed into a CHW program with high degrees of acceptability, adoption, and fidelity.

2. Adoption and fidelity of the surveillance function of the CHEST App by program coordinators was not achieved.

3. The CHEST App intervention is associated with improvements in knowledge of healthy child rearing practices by caregivers.

The improved knowledge scores by the caregivers are theorized to contribute to a reduction in anemia rates and improve ECD scores, with more time passed and higher dosage of the intervention. The CHEST App is a promising tool to improve the performance of CHWs during their home visits, to accomplish their objective of teaching caregivers healthy childrearing practices and improving childhood health and development in their communities.

## Supporting information

**S1 Fig. Theory of change of CHEST App intervention.**
(TIFF)

**S2 Fig. Conceptual model of intervention.**
(TIFF)

**S3 Fig. Implementation research logic model of the CHEST App pilot study.**
(TIFF)

**S4 Fig. SPIRIT checklist reporting guideline.**
(TIF)

**S5 Fig. CONSORT flow diagram.**
(TIF)

**S6 Fig. Trial protocol.**
(TIF)

# Acknowledgments

Special thanks to the team at Elementos whom helped develop the program material and implement the program in the field; Mayra Young, Liz Franco Calderòn, Milagros Alvarado Llatance, and Gabriela Palacios Rojo. Special thanks to the community health workers, program coordinators and municipality workers in Indiana, Mazan, and Fernando Lores, that work so hard for the betterment of the children in their communities.

# Author Contributions

**Conceptualization:** Christopher M. Westgard.

**Data curation:** Christopher M. Westgard.

**Formal analysis:** Christopher M. Westgard, Luis A. Orrego-Ferreyros.

**Funding acquisition:** Christopher M. Westgard.

**Investigation:** Christopher M. Westgard.

**Methodology:** Christopher M. Westgard, Luis A. Orrego-Ferreyros.

**Project administration:** Christopher M. Westgard.

**Supervision:** Christopher M. Westgard.

**Writing – original draft:** Christopher M. Westgard.

**Writing – review & editing:** Christopher M. Westgard, Luis A. Orrego-Ferreyros.

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
