## [Decision Letter · Decision Letter 0]

1 Jul 2022

PGPH-D-22-00792

An mHealth Tool for Community Health Workers to Improve Caregiver Knowledge of Child Health in the Amazon: An Effectiveness-Implementation Hybrid Evaluation

Dear Dr. Westgard,

Thank you for submitting your manuscript to PLOS Global Public Health. After careful consideration, we feel that it has merit but does not fully meet PLOS Global Public Health’s publication criteria as it currently stands. Therefore, we invite you to submit a revised version of the manuscript that addresses the points raised during the review process.

We look forward to receiving your revised manuscript.

Kind regards,

Catherine Elizabeth Draper

Academic Editor

Journal Requirements:

1. You indicated that you had ethical approval for your study. In your Methods section, please ensure you have also stated whether you obtained consent from parents or guardians of the minors included in the study or whether the research ethics committee or IRB specifically waived the need for their consent.

a. Please clarify all sources of funding (financial or material support) for your study. List the grants (with grant number) or organizations (with url) that supported your study, including funding received from your institution. 

b. State the initials, alongside each funding source, of each author to receive each grant.

c. State what role the funders took in the study. If the funders had no role in your study, please state: “The funders had no role in study design, data collection and analysis, decision to publish, or preparation of the manuscript.”

3. Please ensure that Funding Information matches with the Financial Disclosure Statement.

4. Please provide separate figure files in .tif or .eps format and removed from the manuscript file.

5. Please note that your Data Availability Statement is currently missing direct link to access each database. If your manuscript is accepted for publication, you will be asked to provide these details on a very short timeline. We therefore suggest that you provide this information now, though we will not hold up the peer review process if you are unable.

Additional Editor Comments (if provided):

Reviewers' comments:

Reviewer's Responses to Questions

**Comments to the Author**

1. Does this manuscript meet PLOS Global Public Health’s publication criteria? Is the manuscript technically sound, and do the data support the conclusions? The manuscript must describe methodologically and ethically rigorous research with conclusions that are appropriately drawn based on the data presented.

Reviewer #1: Partly

Reviewer #2: Yes

Reviewer #3: Yes

2. Has the statistical analysis been performed appropriately and rigorously?

Reviewer #1: No

Reviewer #2: Yes

Reviewer #3: Yes

3. Have the authors made all data underlying the findings in their manuscript fully available (please refer to the Data Availability Statement at the start of the manuscript PDF file)?

Reviewer #1: Yes

Reviewer #2: Yes

Reviewer #3: Yes

4. Is the manuscript presented in an intelligible fashion and written in standard English?

Reviewer #1: No

Reviewer #2: Yes

Reviewer #3: Yes

5. Review Comments to the Author

Reviewer #1: • The first paragraph is very vague and provides no idea about what this manuscript is about.

• The authors discuss about CHWs role in conducting health promotion and education. However, it is unclear if these practices are geared toward a certain goal, or generic in nature.

• The authors state “ICT and mHealth technology have also been shown to be effective at improving health behavior” – unclear about what this behavior is – nutrition, physical activity, routine management, self-care, quality of life – there can be several different areas.

• In lines 69-70, authors report: “The current study reports on the implementation and intermediary effects of an mHealth innovation to improve CHW performance in the Amazon of Peru.” Unclear presentation of information regarding what the performance is about. What is the outcome of interest?

• The allures to use of implementation strategies. Discussing what the outcome is and why knowledge or evidence about this area is limited will strengthen the background section. If this is a broad area, then providing a description what it entails is required.

• The paper does not really follow a proper methodology reporting. This makes it difficult to gather information. It would be helpful to have a section about design, participants, intervention, instrument, procedure, analysis. The current draft is a mishmash of all of these sections in various combination.

• The authors refer to a previously published paper for details about methodology. However, some details in this paper are required to understand the results and discussion sections.

• The authors didn’t really provide any detail about the CHEST App. As a reader, I am not sure what kind of information was included. The authors report that a video of the app and the animated videos can be found online. However, for the purposes of this manuscript, some details about the content of this app are required to understand what the intervention is about. Without this information, the intervention is only “an app”.

• The results are reported in percentages. Not knowing what the total n is makes it difficult to understand what “13%” or “3%” is.

• The authors provided p-values for comparative data. However, the authors have not reported if the data within each group were normally distributed allowing them to use parametric statistic.

• How were missing data handled?

• Also, for ANOVA results, the authors reported df1 as 2. However, they only reported two treatment levels – intervention and control. This would make df1 = 1. More clarity needed.

• The manuscript will benefit from a thorough edit for English language and grammar.

Reviewer #2: The authors examine implementation of an mHealth tool designed to support CHWs in delivering home-based nurturing care services for caregivers and children in rural Peru. The topic is important, as a growing body of evidence suggests that implementation factors are key in determining the effectiveness of CHW home visit interventions that aim to improve child development in low resource settings. The analytic approach is straightforward, and the manuscript is clearly written. I have several comments:

1. The role of CHW supervision has been shown to be very important in determining the effectiveness of home visit interventions in improving child development outcomes. While the mHealth tool is framed as an aid for content delivery, it would be good to better understand its role in the supervision structure. Is there a function in the tool that is meant to strengthen supervision? Is there a supervisor who regularly checks whether CHWs are using the tool? The role of the “program coordinator” within the supervision structure is not clear.

2. The dosage outcome deserves more attention. A key challenge in scaling home visit interventions for child development is ensuring regular home visits occur, which relates in part to the supervision issue mentioned above, as well as the overall way that the CHW workforce is managed and financed (as evidenced by two municipalities in the study area cancelling their CHW programs entirely). That the number of home visits did not reach the intended goal is consistent with experience in other settings. In general, CHWs are often asked to do too much and skipping visits to the homes of ostensibly healthy children is a common way to manage workload. Do the authors have evidence that the reason for the shortfall was due to COVID-19 and not due to other more common factors that could undermine the potential scale-up of the tool? Is there in a function in the tool designed to support achieving the intended dosage, e.g., providing supervisors with an easy check on the number of home visits a CHW conducts? In general, the success of the tool during the home visits that do occur is an important finding, but if a large percentage of home visits do not occur it is difficult to conclude that the tool is an effective and scalable intervention.

3. Part of the pathway to impact of CHW home visits on child development outcomes could be through direct interactions between CHWs and children themselves. Are there any aspects of the tool that are targeted at children (and any corresponding data on related implementation factors) or is the tool meant entirely to facilitate information sharing between the CHW and caregiver?

4. I suggest a slight reframing of the first paragraph to acknowledge that poverty is the root cause of many of the adverse experiences that are detrimental to child development in settings such as this, even if the more proximal causes include “unhealthy practices in the household”. Poverty severely constrains caregivers’ ability to adopt healthy practices irrespective of knowledge.

Reviewer #3: Overall, the authors did an excellent job in developing the manuscript. Authors report on evaluating an mHealth system (App) implemented to improve CHWs performance in the Amazon of Peru. The manuscript significantly contributes to literature on mHealth innovations in improving healthcare. However, I believe a few items should be addressed before this article is ready for publication. Of particular note, a few grammar issues such as long sentences and grammar errors need to be addressed. Authors should carefully read through the manuscript and correct long sentences that may make it hard for laypersons to read, understand, and appreciate the paper. Authors should also carefully format and structure the manuscript to meet Plos Global Public Health's publication style.

A few additional comments/suggestions that I have are as follows:

1. The study's main aim is not well stated in the background of the abstract and the manuscript's main text. The sample size is not stated in the abstract of the study.

2. The methodology of the manuscript should be well described and structured to clearly highlight the significant components of the methodology of a manuscript. The method of analysis is not well described in the methodology section of the main manuscript. Authors should clearly describe in detail the methods of analysis.

3. Line 345 “affective” should be changed to “effective.”

4. The discussion section of the manuscript section should be well described. Authors should clearly describe how similar or different their findings are from literature. It should describe the social and policy implications of the study findings.

6. PLOS authors have the option to publish the peer review history of their article (what does this mean?). If published, this will include your full peer review and any attached files.

**Do you want your identity to be public for this peer review?** For information about this choice, including consent withdrawal, please see our Privacy Policy.

Reviewer #1: No

Reviewer #2: No

Reviewer #3: No

---

## [Decision Letter · Decision Letter 1]

12 Aug 2022

PGPH-D-22-00792R1

An mHealth Tool for Community Health Workers to Improve Caregiver Knowledge of Child Health in the Amazon: An Effectiveness-Implementation Hybrid Evaluation

Dear Dr. Westgard,

Thank you for submitting your manuscript to PLOS Global Public Health. After careful consideration, we feel that it has merit but does not fully meet PLOS Global Public Health’s publication criteria as it currently stands. Therefore, we invite you to submit a revised version of the manuscript that addresses the points raised during the review process.

We look forward to receiving your revised manuscript.

Kind regards,

Catherine Elizabeth Draper

Academic Editor

Journal Requirements:

1. "Appendix 6. Trial Protocol.tif" file(s) is over our file size limit of 10MB. Please reduce the file size to no more than 10MB. For further help on compressing figures visit: https://journals.plos.org/globalpublichealth/s/figures

Additional Editor Comments (if provided):

I understand that the reviewers' comments were attended to, but I would recommend that some attention is given the additional comments from reviewer 3 to help further enhance the quality of the manuscript.

Reviewers' comments:

Reviewer's Responses to Questions

**Comments to the Author**

1. If the authors have adequately addressed your comments raised in a previous round of review and you feel that this manuscript is now acceptable for publication, you may indicate that here to bypass the “Comments to the Author” section, enter your conflict of interest statement in the “Confidential to Editor” section, and submit your "Accept" recommendation.

Reviewer #2: All comments have been addressed

Reviewer #3: (No Response)

2. Does this manuscript meet PLOS Global Public Health’s publication criteria? Is the manuscript technically sound, and do the data support the conclusions? The manuscript must describe methodologically and ethically rigorous research with conclusions that are appropriately drawn based on the data presented.

Reviewer #2: Yes

Reviewer #3: Yes

3. Has the statistical analysis been performed appropriately and rigorously?

Reviewer #2: Yes

Reviewer #3: Yes

4. Have the authors made all data underlying the findings in their manuscript fully available (please refer to the Data Availability Statement at the start of the manuscript PDF file)?

Reviewer #2: Yes

Reviewer #3: Yes

5. Is the manuscript presented in an intelligible fashion and written in standard English?

Reviewer #2: Yes

Reviewer #3: Yes

6. Review Comments to the Author

Reviewer #2: (No Response)

Reviewer #3: The authors have failed to respond to most of the critical issues raised by reviewers in the first review concerning the background, methodology, and the results and discussion sections of the manuscript. The significance of this manuscript in enhancing global knowledge of the contribution of mHealth to improving child health cannot be overemphasized. Therefore, authors should carefully revisit the reviewer comments from the first review process to enhance the quality and readability of the paper. Below are some few additional comments:

-Overall, the manuscript is not well formatted. There are issues with paragraphing and paragraph spacing.

-The background section is still vague and unclear. It should be clearly written and well structured to demonstrate a clear overview of the study. The main purpose of the study should also be stated clearly in the background. For instance, lines 67-68 and lines 73-77 tries to highlight the purpose of the study. However, it is not very clearly stated.

-Line 79-108 can be moved to the methods section where the study intervention was described.

-The methodology section should be structured to follow a proper research methodology reporting format. Authors should describe each method section clearly to enhance understanding and readability.

7. PLOS authors have the option to publish the peer review history of their article (what does this mean?). If published, this will include your full peer review and any attached files.

**Do you want your identity to be public for this peer review?** For information about this choice, including consent withdrawal, please see our Privacy Policy.

Reviewer #2: No

Reviewer #3: No

---

## [Decision Letter · Decision Letter 2]

2 Sep 2022

An mHealth Tool for Community Health Workers to Improve Caregiver Knowledge of Child Health in the Amazon: An Effectiveness-Implementation Hybrid Evaluation

PGPH-D-22-00792R2

Dear Westgard,

We are pleased to inform you that your manuscript 'An mHealth Tool for Community Health Workers to Improve Caregiver Knowledge of Child Health in the Amazon: An Effectiveness-Implementation Hybrid Evaluation' has been provisionally accepted for publication in PLOS Global Public Health.

Best regards,

Zulkarnain Jaafar

Academic Editor

Reviewer Comments (if any, and for reference):

Reviewer's Responses to Questions

**Comments to the Author**

1. If the authors have adequately addressed your comments raised in a previous round of review and you feel that this manuscript is now acceptable for publication, you may indicate that here to bypass the “Comments to the Author” section, enter your conflict of interest statement in the “Confidential to Editor” section, and submit your "Accept" recommendation.

Reviewer #2: All comments have been addressed

Reviewer #3: All comments have been addressed

2. Does this manuscript meet PLOS Global Public Health’s publication criteria? Is the manuscript technically sound, and do the data support the conclusions? The manuscript must describe methodologically and ethically rigorous research with conclusions that are appropriately drawn based on the data presented.

Reviewer #2: Yes

Reviewer #3: Yes

3. Has the statistical analysis been performed appropriately and rigorously?

Reviewer #2: Yes

Reviewer #3: Yes

4. Have the authors made all data underlying the findings in their manuscript fully available (please refer to the Data Availability Statement at the start of the manuscript PDF file)?

Reviewer #2: Yes

Reviewer #3: Yes

5. Is the manuscript presented in an intelligible fashion and written in standard English?

Reviewer #2: Yes

Reviewer #3: Yes

6. Review Comments to the Author

Reviewer #2: (No Response)

Reviewer #3: (No Response)

7. PLOS authors have the option to publish the peer review history of their article (what does this mean?). If published, this will include your full peer review and any attached files.

**Do you want your identity to be public for this peer review?** For information about this choice, including consent withdrawal, please see our Privacy Policy.

Reviewer #2: No

Reviewer #3: No
